

# Ice nucleating particles in Canadian Arctic sea-surface microlayer and bulk seawater

Victoria E. Irish[1], Pablo Elizondo[1], Jessie Chen[1], Cédric Chou[1], Joannie Charette[2], Martine Lizotte[3], Luis A. Ladino[4*], Theodore W. Wilson[5], Michel Gosselin[2], Benjamin J. Murray[5], Elena Polishchuk[1], Jonathan P. D. Abbatt[4], Lisa A. Miller[6], Allan K. Bertram[1]

[1] Department of Chemistry, University of British Columbia, 2036 Main Mall, Vancouver, BC V6T 1Z1, Canada
[2] Institut des sciences de la mer de Rimouski, Université du Québec à Rimouski, 310 Allée des Ursulines, Rimouski, Québec, QC G5L 3A1, Canada
[3] Département de biologie, Québec-Océan, Université Laval, Québec, QC G1V 0A6, Canada
[4] Department of Chemistry, University of Toronto, 80 St George Street, Toronto, Ontario, ON M5S 3H6, Canada
[*] Current address is: Centro de Ciencias de la Atmósfera, Universidad Nacional Autónoma de México, Ciudad Universitaria, Mexico City, Mexico
[5] Institute for Climate and Atmospheric Science, School of Earth and Environment, University of Leeds, Woodhouse Lane, Leeds, LS2 9JT, UK
[6] Institute of Ocean Sciences, Fisheries and Oceans Canada, Sidney, BC V8L 4B2, Canada
*Correspondence to: Allan Bertram (bertram@chem.ubc.ca)*

**Abstract:**

The sea-surface microlayer and bulk seawater can contain ice-nucleating particles (INPs) and these INPs can be emitted into the atmosphere. Our current understanding of the properties, concentrations, spatial and temporal distributions of INPs in the microlayer and bulk seawater is limited. In this study we investigate the concentrations and properties of INPs in microlayer and bulk seawater samples collected in the Canadian Arctic during the summer of 2014. INPs were ubiquitous in the microlayer and bulk seawater with freezing temperatures as high as -14 °C. A strong negative correlation (R = -0.7, p = 0.02) was observed between salinity and freezing temperatures (after correction for freezing depression by the salts). One possible explanation is that INPs were associated with melting sea ice. Heat and filtration treatments of the samples show that the INPs were likely biological materials with sizes between 0.02 μm and 0.2 μm in diameter, consistent with previous measurements off the coast of North America and near Greenland in the Arctic. The concentrations of INPs in the microlayer and bulk seawater were consistent with previous measurements at several other locations off the coast of North America. However, our average microlayer concentration was lower than previous observations made near Greenland in the Arctic. This difference could not be explained by chlorophyll *a* concentrations derived from satellite measurements. In addition, previous studies found significant INP enrichment in the microlayer, relative to bulk seawater, which we did not observe in this study. While further studies are needed to understand these differences, we confirm that there is a source of INP in the microlayer and bulk seawater in the Canadian Arctic that may be important for atmospheric INP concentrations.

## 1 Introduction

Ice can form in clouds by homogeneous or heterogeneous ice nucleation. Homogeneous ice nucleation refers to ice nucleation in the absence of a foreign substrate, while heterogeneous ice nucleation refers to ice nucleation initiated by a foreign substrate or an ice-nucleating particle (INP). Homogeneous ice nucleation becomes increasingly important below approximately -33 °C, but INPs can trigger ice formation in clouds at higher temperatures (Herbert et al., 2015; Koop and Murray, 2016). Therefore INPs in the atmosphere can affect Earth's climate and the hydrological cycle by altering the microphysics, radiative properties, and lifetime of clouds (DeMott et al., 2010; Lohmann, 2002; Lohmann and Feichter, 2005; Tan et al., 2016).





Field and laboratory studies have shown that the sea-surface microlayer and bulk seawater contain INPs, and that these INPs can be emitted to the atmosphere by the bubble bursting mechanism (Alpert et al., 2011a, 2011b; Blanchard, 1964; DeMott et al., 2015; Fahlgren et al., 2015; Fall and Schnell, 1985; Knopf and Forrester, 2011; Prather et al., 2013; Rosinski et al., 1988; Schnell, 1977; Schnell and Vali, 1975, 1976; Vali et al., 1976; Wang et al., 2015; Wilson et al., 2015). The sea-surface microlayer (herein referred to as the microlayer) is the interface between the ocean and the atmosphere. The thickness of the microlayer is < 1 mm (Liss and Duce, 1997), and the physical and chemical properties of the microlayer are different from the bulk seawater (Zhang et al., 2003). For example, the concentration of organic material is often enhanced in the microlayer compared to the bulk seawater (Wurl et al., 2009).

Modelling studies have suggested that the ocean can be a dominant source of INPs in the atmosphere when dust concentrations are low (Burrows et al., 2013; Vergara-Temprado et al., 2017; Wilson et al., 2015). Modelling studies have also suggested that marine INPs may offset the magnitude of anthropogenic aerosol forcing by influencing cloud formation (Yun and Penner, 2013). Nevertheless, our current understanding of the properties, concentrations, and spatial and temporal distributions of INPs in the microlayer and bulk seawater remains limited, leading to uncertainties when predicting their transfer towards the atmosphere and their impacts on climate and the hydrological cycle.

Prior to our work, five studies had examined INPs in bulk waters around North America and near Greenland (Fig. 1), but only one quantified INPs in the microlayer in the immersion mode (Wilson et al., 2015). The immersion mode refers to heterogeneous freezing caused by INPs immersed in liquid droplets, which is the mode most relevant for mixed-phase clouds in the atmosphere (Murray et al., 2012). Our work adds more measurements to the limited data on INPs in the microlayer and bulk seawater, contributing to a better understanding of how the properties and concentrations of INPs in the microlayer vary with location and time.

We investigated the concentrations and properties of INPs in the microlayer and bulk seawater samples in the immersion mode collected in the Canadian Arctic (Fig. 1) during the summer of 2014. The Arctic was chosen for these studies because 1) clouds in this region have been found to be especially sensitive to atmospheric concentrations of INPs (Harrington et al., 1999; Jiang et al., 2000), 2) there have not been previous studies of the freezing properties of the microlayer or bulk seawater in this region, and 3) as sea ice continues to decrease in the Arctic, the microlayer and bulk seawater may become more important sources of INPs in this region.

## 2 Experimental

### 2.1 Sampling locations and collection methods

All samples were collected during July and August 2014 from the eastern Canadian Arctic on board the Canadian research icebreaker CCGS *Amundsen* as part of the Network on Climate and Aerosols: addressing key uncertainties in Remote Canadian Environments (NETCARE) project. The locations of the eight stations sampled in this study are shown in Fig. 1 while Table 1 describes sampling times and specific geographic coordinates of these stations. Supplementary details, including notes and photographs taken at each station during sampling are provided in Table S1.

The microlayer samples were collected using a glass plate sampler (Harvey and Burzell, 1972) from the upwind side of a small inflatable, rigid-hull boat, at least 500 m away from the CCGS *Amundsen* to avoid contamination. The glass plate was immersed vertically and withdrawn at a slow rate (between 3 to 5 cm/s) and allowed to drain for less than 5 s. The microlayer that adhered to the plate from each dip was scrapped off from one side of the glass plate with a neoprene wiper blade into a 1L high-density polyethylene (HDPE) Nalgene bottle. For each microlayer sample, approximately 500 – 1000 mL was collected, requiring 115-



185 dips. Based on the amount of material collected, the number of dips and the area of the plate, the thickness of the layer collected ranged between 60 and 220 μm. Bulk seawater samples were collected at the same times and locations as the microlayer samples using a Niskin bottle deployed from the downwind side of the zodiac. Samples were collected at 0.5 m depth and transferred to 1L HDPE Nalgene bottles. After collection, the Nalgene bottles containing both the microlayer and bulk

samples were kept cool in an insulated container. Upon returning to the ship, the samples were homogenised by gently inverting them at least ten times and then sub-sampled into smaller bottles for subsequent analyses.

The glass plate, neoprene wiper blade and all Nalgene bottles were cleaned with isopropanol and ultra pure water and rinsed with approximately 10 mL of the seawater sample before use. The Niskin bottle was not cleaned with isopropanol before sampling, but the inside of the bottle was rinsed with a large amount of seawater before sample collection.

**2.2 Ice nucleation properties of the samples**

**2.2.1 Droplet freezing technique and INP concentrations**

INP concentrations as a function of temperature were determined using the droplet freezing technique (DFT; Koop et al., 1998; Vali, 1971; Whale et al., 2015; Wilson et al., 2015). Sub-samples of the microlayer and bulk seawater were stored in Nalgene bottles frozen at -80 °C for a maximum of nine months before INP analysis. A previous study suggests that freezing

seawater samples does not significantly change the freezing properties of the samples (Schnell and Vali, 1975). Each microlayer and bulk seawater sample was completely thawed and homogenised by inverting at least ten times. Between 15 to 20 droplets of the sample, with volumes of 0.6 μL each, were deposited onto a hydrophobic glass slide (HR3-215; Hampton Research, Aliso Viejo, CA, USA) using a pipette. The slides were put into an airtight cell (Parsons et al., 2004), attached to a cold stage and analysed by the DFT as detailed in Wheeler et al. (2015). The droplets were cooled at a constant rate of 5 °C/min from 0 °C to -

35 °C. Each experiment was repeated three times using three different slides. Blanks were determined by filtering the microlayer and bulk samples through a 0.02 μm Anotop 25 filter. Ultrapure water (distilled water further purified with a Millipore system) was also analysed for INPs using the DFT for comparison.

The concentrations of INPs per volume of sample, *[INP]*, was determined from each freezing experiment by the following equation (Vali, 1971):

$$\left[ INP \right] = -\ln\left(\frac{N_u(T)}{N_o}\right) N_o \cdot \frac{1}{V} \tag{1}$$

Where $N_u(T)$ is the number of unfrozen droplets at temperature $T$, $N_o$ is the total number of droplets used in the experiment and $V$ is the volume of all droplets in a single experiment. This equation accounts for the possibility of multiple INPs contained in a single droplet.

**2.2.2 Heating tests**

The freezing temperatures of the microlayer and bulk samples were also measured after they had been heated to 100 °C (Christner et al., 2008; Schnell and Vali, 1975; Wilson et al., 2015). This temperature was chosen because some biological materials have been shown to lose their ice nucleation activity following heating to 95 °C (Christner et al., 2008), possibly due to denaturation of the tertiary structure of ice nucleating proteins (Hill et al., 2016). Samples of microlayer and bulk seawater were put into polypropylene tubes, sealed with lids, and heated to 100 °C in a heating block (Accublock, Labnet, S/N: D1200) for an

hour, then cooled to room temperature for approximately 30 minutes before freezing measurements.





### 2.2.3 The size of the INPs

Following Wilson et al. (2015), the microlayer and bulk seawater samples were passed through filters with three different pore sizes (Whatman 10 μm pore size PTFE membranes, Millex –HV 0.2 μm pore size PTFE membranes, and Anotop 25 0.02 μm pore size inorganic Anopore™ membranes). These filtered samples were subsequently used in the freezing
measurements.

### 2.2.4 Corrections for freezing temperature depression

Since the microlayer and bulk seawater samples contained salts, the measured freezing temperatures were adjusted for the presence of the salts. Using measured salinities and the water activity based approach (Koop and Zobrist, 2009) hypothetical heterogeneous freezing temperatures for salt-free conditions were obtained (salinity = 0 g/kg). For details see Supplement,
Section 1. The salinities of the microlayer and bulk seawater samples were measured within 10 minutes of sample collection using a hand-held salinity probe (SympHony; VWR, Radnor, PA, USA) which had been calibrated against discrete seawater samples analysed on a Guildline Autosal 8400B. The correction for the presence of salts based on the measured salinities ranged from 2.0 to 2.8 °C. Hypothetical heterogeneous freezing temperatures for salt-free conditions is more relevant for mixed phase clouds, where freezing typically occurs in dilute aqueous droplets with low salt concentrations (i.e., where water activity tends
toward unity).

### 2.3 Phytoplankton and bacterial abundance

Duplicate 5 mL sub-samples were fixed with 20 μL of 25% Grade I glutaraldehyde (0.1% final concentration; Sigma-Aldrich G5882) and kept frozen at -80 °C until analysis by flow cytometry, within 7 months of collection (Marie et al., 2005). Cyanobacteria were identified by orange fluorescence from phycoerythrin (575 ± 20 nm). Heterotrophic bacteria samples were
stained with SYBR Green I and measured at 525 nm to detect low and high nucleic acid content (Belzile et al., 2008). Archaea could not be discriminated from bacteria using this protocol; therefore, hereafter, we use the term bacteria to include both archaea and bacteria. Photosynthetic eukaryotes were identified by red fluorescence of chlorophyll (675 ± 10 nm). In each sub-sample, microspheres (1 μm and 2 μm, Fluoresbrite plain YG, Polysciences) were added as an internal standard as described by Tremblay et al. (2009). Analyses were performed on an Epics Altra flow cytometer (Beckman Coulter), fitted with a 488 nm
laser (15 mW output; blue), using Expo32 v1.2b software (Beckman Coulter).

### 2.4 Dimethlysulphide (DMS) measurements

Concentrations of DMS were measured on board the ship within approximately two hours of sampling. The samples were analysed by gas chromatography following purging and cryo-trapping according to protocol described in Lizotte et al. (2008).

### 30 2.5 Statistical analysis

Pearson correlation analysis was applied to many of the variables measured in this study to compute correlation coefficients (R). Here we use the scheme from Dancey and Reidy (2002) where correlations with R values of 0.1-0.3, 0.4-0.6 and 0.7-0.9 are classified as weak, moderate and strong, respectively. P values were also calculated to determine if the correlations were statistically significant at the 95 % confidence level (p < 0.05).



### 3 Results and Discussion

#### 3.1 INPs in the microlayer and bulk seawater

The fraction of droplets frozen in the immersion mode for both the unfiltered microlayer and bulk seawater samples are shown in Fig. 2. In this figure the blanks refer to the freezing properties of the sample after 0.02 μm filtration. The freezing

properties of the blanks (after correction for freezing point depression by the salts) are similar to or lower than the freezing properties of ultrapure water, which are also shown in Fig. 2. The fraction-frozen curves for each station fall at warmer temperatures than their respective blanks, indicating that the microlayer and bulk seawater samples from all stations contained INPs. Box plots of the $T_{10}$-values for the blanks, the microlayer and bulk seawater samples are shown in Fig. 3, where $T_{10}$ represents the temperatures at which 10% of droplets had frozen. Figure 3 shows that the interquartile range of freezing

temperatures for the samples is higher than the interquartile range of freezing temperatures for the blanks, further illustrating that INPs were present in the microlayer and bulk seawater samples.

The freezing curves varied significantly from sample to sample (Fig. 2). To understand this variability we investigated correlations between the $T_{10}$-values for the bulk seawater samples and chemical and physical properties of the bulk seawater (DMS concentration, bacterial and phytoplankton abundance, seawater temperature, pH and salinity). Correlation coefficients

were not statistically significant ($p > 0.05$), except in the case of salinity (Table 2). A strong negative correlation ($R = -0.7$, $p = 0.02$) was observed between salinity and the $T_{10}$-values (corrected for freezing depression by the salts). This suggests that more INPs were found in less saline waters. One possible explanation is that the INPs were associated with melting sea ice. Materials such as algal aggregates, sea ice diatoms and extracellular polymeric substances can be released into the ocean upon sea ice melting (Assmy et al., 2013; Boetius et al., 2015; Fernández-Méndez et al., 2014) and might be potential sources of the INPs

observed in this study .

The concentration of INPs, *[INP]*, as a function of temperature for the microlayer samples analysed in this study is shown in Fig. 4A. Also included in Fig. 4A are results from Wilson et al. (2015) for the microlayer samples they collected at the locations shown in Fig. 1A. Concentrations of INPs in microlayer samples at stations 2, 9, and 10 overlap with the INP concentrations observed by Wilson et al. (2015) in the Atlantic. However, the INP concentrations in the microlayer measured by Wilson et al.

(2015) to the east of Greenland are higher than the concentrations measured here.

Figure 4B shows the concentrations of INPs as a function of temperature for the bulk seawater samples. Also included in Fig. 4B are results from other studies (see Fig. 1B for locations) that measured INPs in samples of bulk seawater or samples containing a mixture of the microlayer and bulk seawater. The range of concentrations observed in our studies agrees well with the range observed by Schnell and Vali (1975), Schnell (1977) and Wilson et al., (2015) (both Arctic and Atlantic). Note, the bulk seawater

freezing data from Wilson et al., (2015) was at the detection limit of their instrument; therefore, their INP concentrations for bulk seawater should be considered upper limits.

A strong positive correlation ($R = 0.9$, $p = 0.002$) between the freezing properties of the microlayer and the freezing properties of the bulk seawater was observed in the current study. Shown in Fig. 5A is a correlation plot between the $T_{10}$-values from the microlayer and bulk seawater samples. The data points, except for one, fall upon the 1:1 line, if the uncertainties in the

measurements are considered. In contrast, Wilson et al. (2015) found significantly more INPs in the microlayer than in bulk seawater (Fig. 5B) in both their Arctic and Atlantic samples. Figure 5 also shows correlation plots for bacterial abundance in the microlayer and bulk seawater for this study (Fig. 5C) and from Wilson et al. (2015) (Fig. 5D). Similar bacterial abundances were observed in the microlayer and bulk seawater in the current study, whereas Wilson et al. (2015) found a higher bacterial abundance in the microlayer compared to the bulk seawater in most samples (Fig. 5D).



The differences between the results in the current study and the results from Wilson et al. (2015) may be, in part, related to sampling techniques. In the current study, the bulk seawater was sampled from a depth of 0.5 m while Wilson et al., (2015) sampled from a depth of 2-5 m. In addition, in the current study the glass plate technique used collected a layer that was up to 220 µm thick, while Wilson et al. (2015) used a hydrophilic Teflon film on a rotating drum fitted to a remote-controlled sampling

catamaran which collects a microlayer of thickness between 6 to 83 µm (Knulst et al., 2003). Other studies have shown that different sampling techniques lead to different measured enrichments of the microlayer. Aller et al. (2017) compared the enrichments of the microlayer determined with the glass plate and a hydrophilic Teflon film on a rotating drum. They observed an enrichment (by a factor of approximately two) of bacteria in the microlayer when using the rotating drum, but no enrichment when using the glass plate technique. In addition, they observed an enrichment of transparent exopolymer material in the

microlayer when using the rotating drum, but a smaller enrichment was observed when using the glass plate technique. Note that Aller et al. (2017) allowed seawater to stand in a 250 gallon tank for one hour before sampling the microlayer with a glass plate whereas the microlayer sampled with the rotating drum was taken directly from the ocean. Additional studies are needed to determine if the methodology used to sample the microlayer and bulk seawater strongly influences measured INP concentrations. The differences between the results in the current study and the results from Wilson et al. (2015) may also be related to

differences in the state of the ocean at the time of sampling. To investigate this we compared monthly average chlorophyll *a* concentrations for both studies. As illustrated in Figs. S1-S3 (Supplement) a clear difference between chlorophyll *a* concentrations in the current study and the Wilson et al. (2015) study was not observed.

Wind speed could also affect the stability of the microlayer and explain differences between results from the current study and the Wilson et al. (2015) study. Previous studies suggest that a microlayer may be stable up to the global average wind speed of

6.6 m/s (Wurl et al., 2011). During the current study, sampling was carried out at wind speeds ranging from 0.7 m/s to 6.7 m/s, while Wilson et al., (2015) carried out sampling at wind speeds ranging from 1.2 m/s to 5.9 m/s. The similar wind speeds in both studies and the fact that almost all sampling was carried out with wind speeds less than the global average suggests that the observed differences in INP concentrations is not due to wind speeds.

### 3.2 Properties of the INPs

### 3.2.1. Heat-sensitive biological material

The fraction frozen curves of samples before and after heating to a temperature of 100 °C are shown in Fig. 6. For 7 out of 8 of the microlayer samples, and all of the bulk samples, the fraction-frozen curves are shifted to colder temperatures after heating. These results suggest that the INPs in most cases are heat-sensitive biological material, consistent with previous measurements of the properties of INPs in the microlayer (Wilson et al., 2015) and bulk seawater (Schnell and Vali, 1975, 1976;

Schnell, 1977).

### 3.2.2 Size of INPs

The $T_{10}$-values as a function of filter pore size (0.02 µm, 0.2 µm and 10 µm) are shown in Fig. 7. For over half the samples (microlayer samples at Stations 4 and 5, bulk samples at Station 6 and bulk and microlayer samples at Stations 7, 9, and 10) the sizes of the INPs were clearly between 0.02 µm and 0.2 µm, as the $T_{10}$-values significantly decreased when the samples

were passed through a 0.02 µm filter but not when passed through a 0.2 µm filter. For the other samples (bulk samples at Stations 4 and 5, microlayer samples at Station 6, and microlayer and bulk samples at Stations 2 and 8), the uncertainties were too large to draw a clear conclusion about the effect of filtration. The 0.02 - 0.2 µm size range for the INPs identified here is consistent with previous studies of INPs in the microlayer or bulk seawater. Wilson et al. (2015) concluded that INPs in the microlayer were



between 0.02 μm and 0.2 μm in size. Rosinski et al. (1986) found that ice freezing nuclei in aerosol of marine origin were below 0.5 μm in size. Schnell and Vali (1975) found ocean-derived ice nuclei to be below 1 μm in size.

The size of whole cell marine bacteria or phytoplankton (excluding femtoplankton) is typically greater than 0.2 μm (Burrows et al., 2013; Sieburth et al., 1978), hence whole cell marine bacteria are unlikely to be the source of the INPs identified here.

Furthermore, correlations between INP concentrations and bacterial or phytoplankton abundance were not statistically significant ($p$-values > 0.05; see supplemental Table S2). This is consistent with the suggestion that whole cells are not the source of the INPs. Potential sources of the INPs observed in this study include ultramicrobacteria, viruses, phytoplankton exudates, or bacteria exudates, all of which could be denatured by heat, but are less than 0.2 μm in size (Ladino et al., 2016; Wilson et al., 2015).

**4 Summary and conclusions**

Concentrations of INPs in the microlayer and bulk seawaters at eight different stations in the Canadian Arctic were determined. Results showed that the INPs were ubiquitous in the microlayer and bulk seawater and that freezing temperatures as high as -14°C were observed in both the microlayer and bulk seawater. A strong negative correlation ($R$ = -0.7, $p$ = 0.02) was observed between salinity and freezing temperatures (after correction for freezing depression by salts). One possible explanation

is that INPs were associated with melting sea ice. The concentration of INPs in the bulk seawater was in good agreement with concentrations observed in bulk samples at several other locations in the Northern Hemisphere. The concentrations of INPs in the microlayer were consistent with concentrations observed by Wilson et al., (2015) off the coast of North America. Heating the samples substantially reduced the INPs' activity, suggesting that biological materials were the likely source of that activity. Filtration of the samples showed that the INPs were between 0.02 μm and 0.2 μm, implying that the ice-active biological

material was likely ultramicrobacteria, viruses, or extracellular material, rather than whole cells.

We conclude that the concentrations and properties of INPs in the microlayer and bulk seawater in the Canadian Arctic are similar to other locations previously studied. However, there were some important differences. On average the concentration of INPs in the microlayer in the current study was lower than the average concentration of INPs measured by Wilson et al., (2015). These differences could not be explained by chlorophyll *a* concentrations from satellite measurements. In addition, similar

concentrations of INPs in the microlayer and bulk seawater were observed here, while Wilson et al., (2015) observed significant enrichment of INPs in the microlayer compared to the bulk seawater. The differences may be related to sampling techniques, but they could also be due to the oceanic state during sampling. Further studies are needed to understand how measured concentrations of INPs in the microlayer and bulk seawater depend on sampling techniques. Further studies are also needed to understand how measured concentrations of INPs in the microlayer and bulk seawater depend on oceanic variables, particularly

changing sea-ice distributions.

As sea ice in the Arctic continues to decrease, the microlayer and bulk seawater could play a larger role in the overall atmospheric INP population in this region. Future modelling studies are needed to determine the magnitude of the effect this INP source has on cloud microphysics in the Arctic region and how it might change as sea-ice distributions change.

**Acknowledgements**

We would like to thank the scientists, officers, and crew of the CCGS *Amundsen* for their support during the expedition; Mélanie Simard and Claude Belzile for help with analysis; and Drs. Dennis A. Hansell and Wenhao Chen for providing Reference Materials. We would also like to thank the Natural Sciences and Engineering Research Council of Canada and Fisheries and



Oceans Canada for funding. BJM acknowledges support from The European Research Council, (ERC 648661 MarineIce) and

the Natural Environment Research Council (NERC, NE/K004417/1).



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





| Station number | Sampling start time (UTC)* | Location |
|---|---|---|
| Station 2 | 23rd July 2014 17:10 | 74°36'935N 94°43'663W |
| Station 4 | 30th July 2014 22:10 | 76°19'882N 071°10'329W |
| Station 5 | 31st July 2014 21:00 | 76°16'568N 074°36'063W |
| Station 6 | 3rd Aug 2014 12:20 | 81°21'743N 064°11'399W |
| Station 7 | 4th Aug 2014 18:40 | 79°58'672N 069°56'051W |
| Station 8 | 5th Aug 2014 19:20 | 79°04'673N 071°39'205W |
| Station 9 | 11th Aug 2014 20:00 | 69°10'009N 100°44'018W |
| Station 10 | 12th Aug 2014 18:50 | 68°55'897N 105°19'809W |

*Sampling took 45-90 minutes to complete.

**Table 1 – Sampling times and geographic coordinates for the eight stations investigated during July-August 2014 in the Canadian Arctic.**



| Chemical and physical properties | $T_{10}$-value | | |
|---|---|---|---|
| | R | p | n |
| Dimethylsulphide concentration | -0.6 | 0.074 | 8 |
| Bacterial abundance | -0.4 | 0.189 | 6 |
| Phytoplankton abundance | -0.5 | 0.138 | 6 |
| Temperature | 0.1 | 0.381 | 8 |
| pH | -0.1 | 0.372 | 8 |
| Salinity | **-0.7** | **0.020** | **8** |

**Table 2 - Correlation analyses between chemical or physical properties of bulk seawater and $T_{10}$-values for the bulk seawater samples. Numbers in bold represent correlations that are statistically significant ($p < 0.05$).**



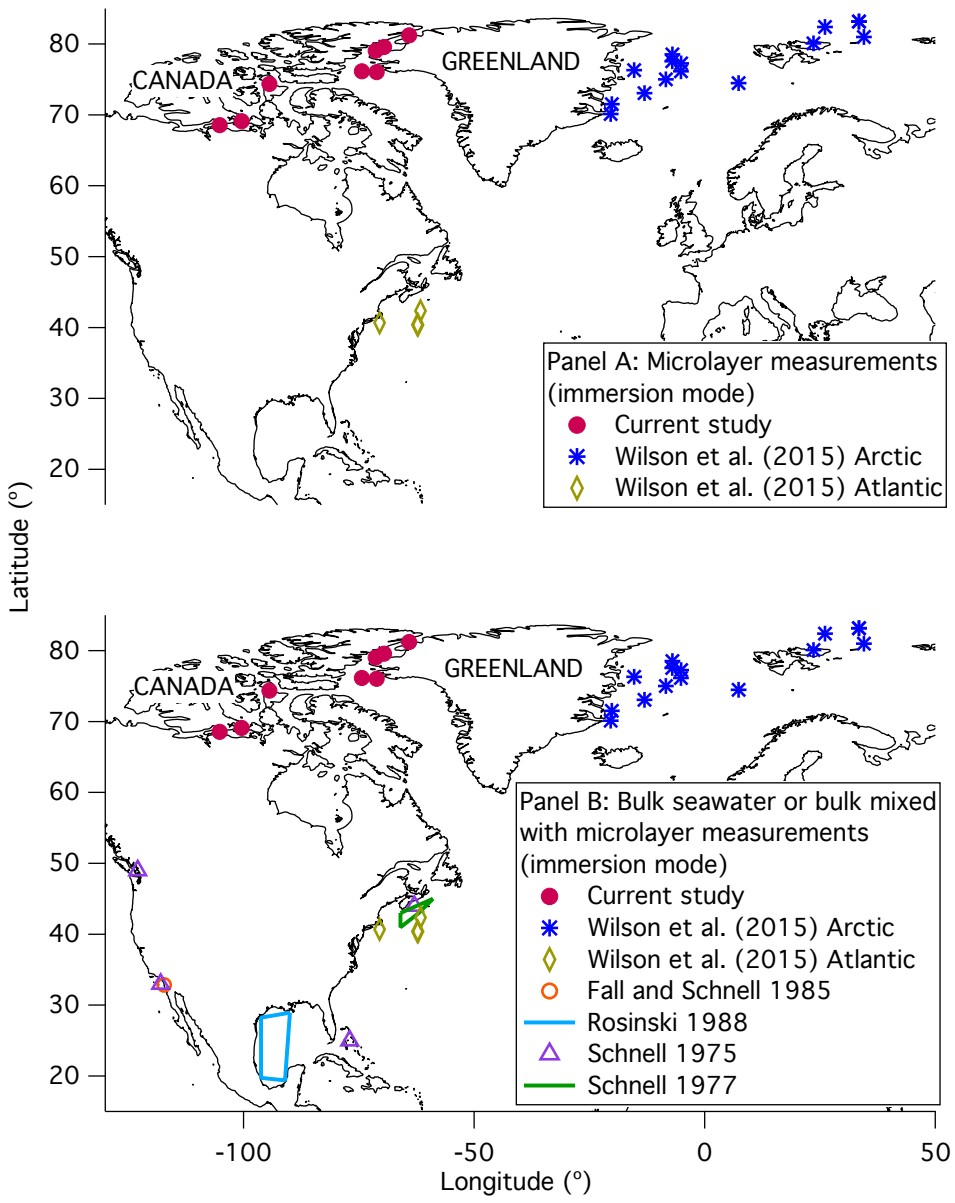

**Figure 1 - Panel A: locations of current and previous studies of INPs (immersion mode) in the microlayer. Panel B: locations of current and previous studies of INPs (immersion mode) in bulk seawater or mixtures of bulk seawater and microlayer. Dates and coordinates for samples in the current study can be found in Table 1.**





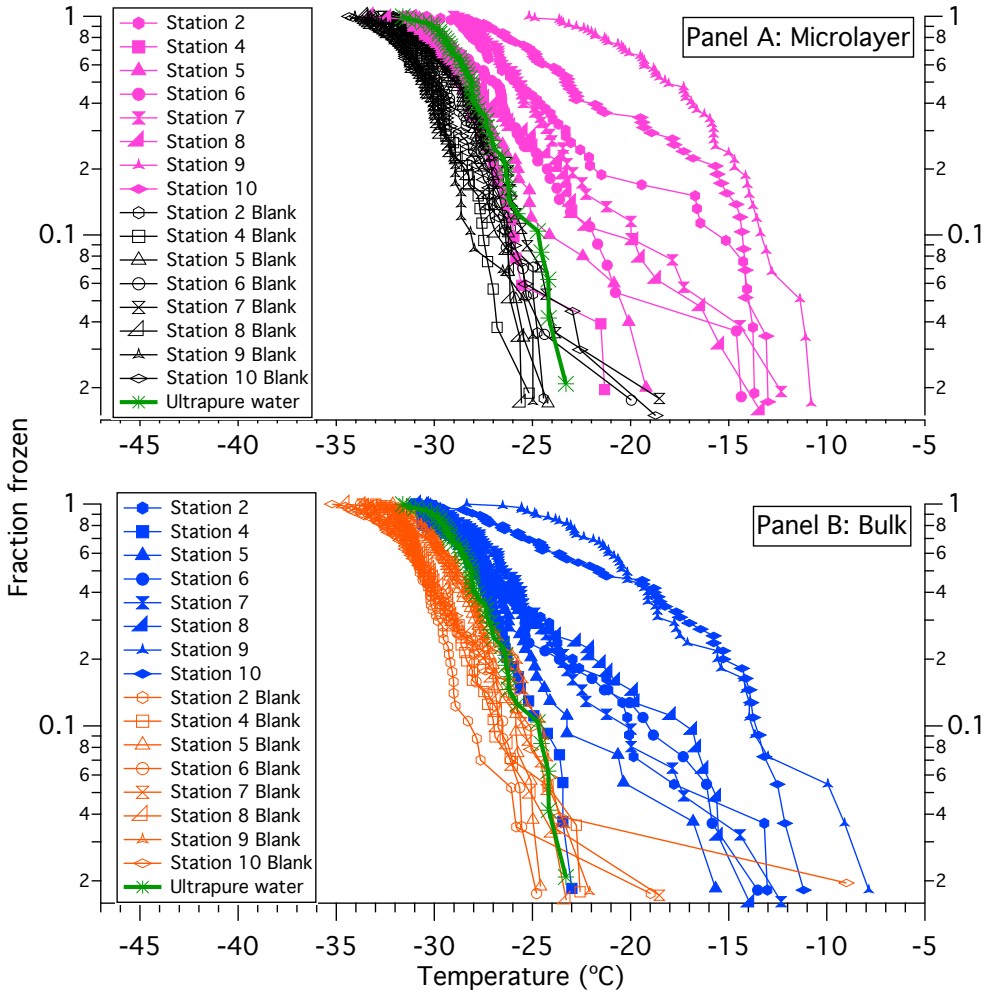

**Figure 2 - Fraction of droplets frozen (in the immersion mode) versus temperature. Panel A and Panel B correspond to the microlayer and bulk seawater, respectively. Also included are the respective blank samples and ultrapure water. Each data point corresponds to a single freezing event in the experiments. All microlayer and bulk seawater freezing points have been corrected for freezing point depression to account for dissolved salts in seawater (Section 2.2.4).**



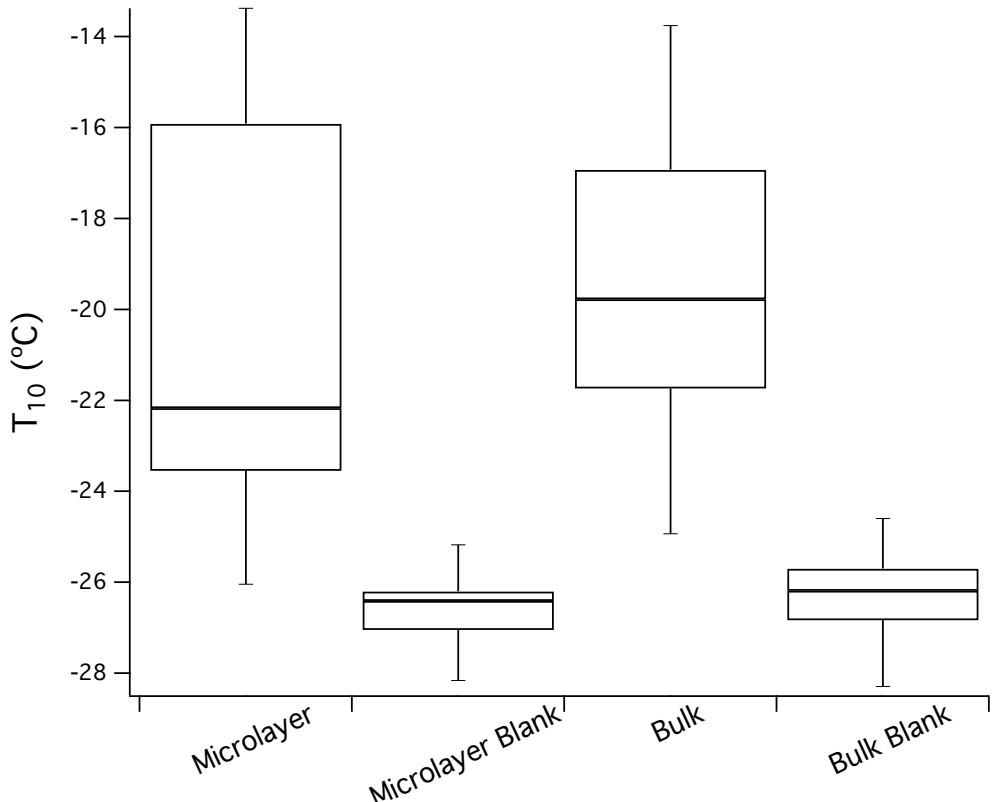

**Figure 3 - Temperature at which 10% of droplets had frozen ($T_{10}$) for microlayer and bulk seawater samples. All data have been corrected for freezing point depression. Boxes represent the 25th, 50th and 75th percentiles, and whiskers represent the minima and maxima.**




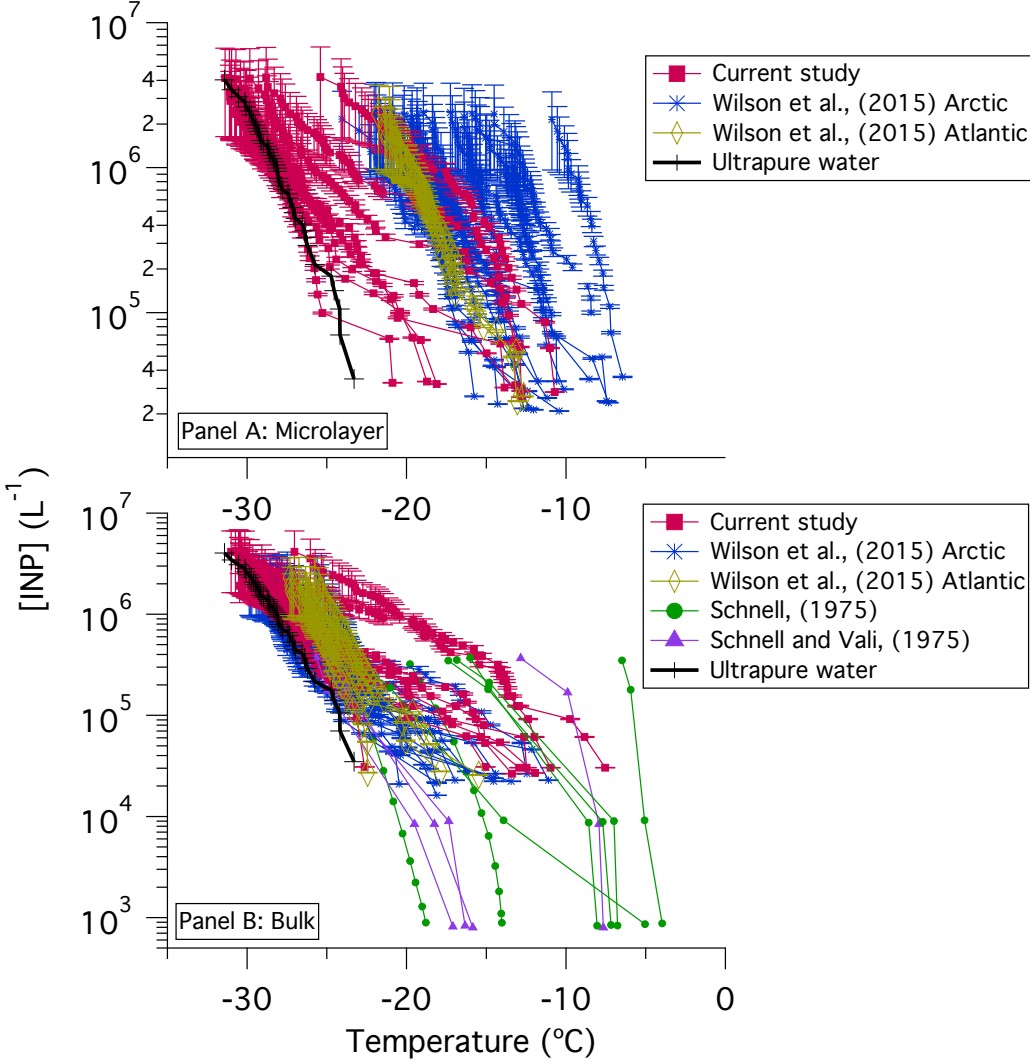

**Figure 4 - The concentrations of INPs per volume of sample [INP] in the microlayer (Panel A) and bulk seawater samples (Panel B). All data, including those from other studies, are corrected for freezing point depression.**




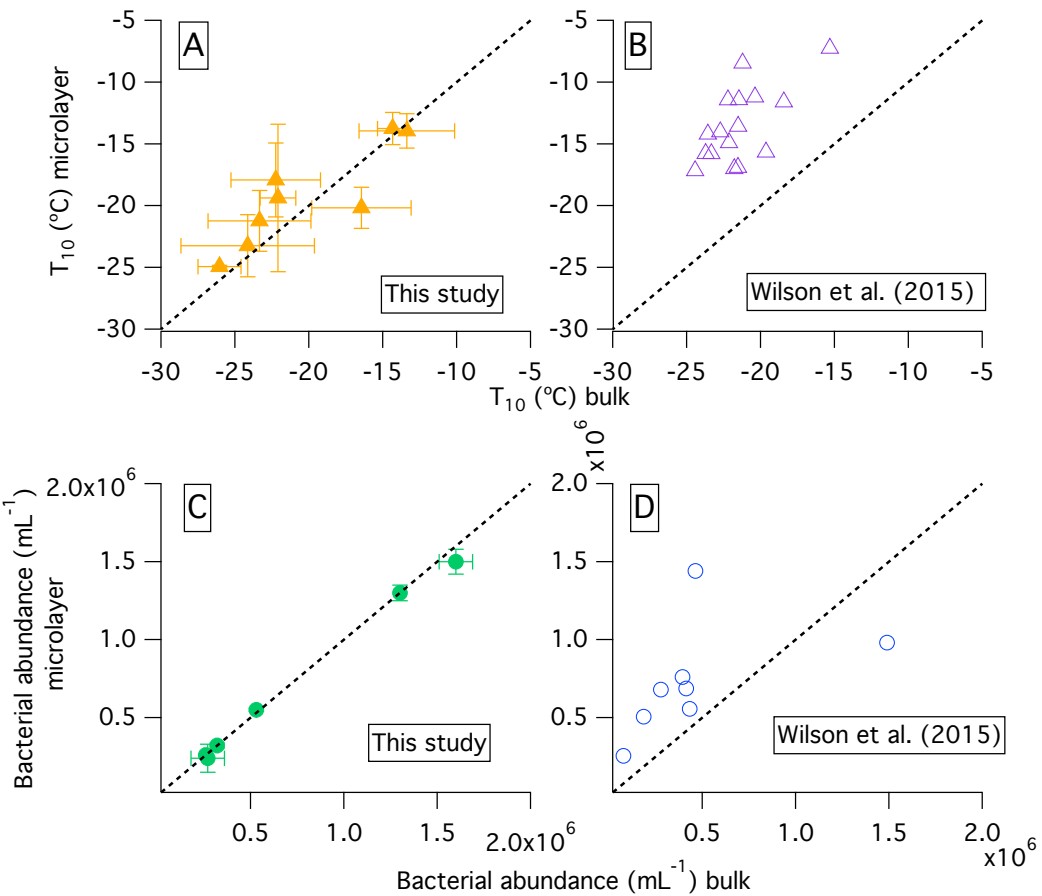

**Figure 5 - Correlation plots with a 1:1 line for reference. Panel A: freezing temperatures for microlayer and bulk seawater samples in this study. $T_{10}$ represents the freezing temperatures at which 10% of the droplets had frozen. All error bars represent the 95% confidence intervals of the $T_{10}$-values from 3 replicate experiments. All data have been corrected for freezing point depression. Panel**
5 **B: $T_{10}$-values for microlayer and bulk seawater samples from Wilson et al. (2015). All data have been corrected for freezing point depression. The reported $T_{10}$ values for their bulk samples should be considered upper limits, since their bulk freezing data were at the detection limit of their instrument. Panel C: bacterial abundance in the microlayer and bacterial abundance in the bulk seawater in this study. There was only one reliable microlayer sample from station 7 for bacterial abundance; therefore, the percentage error for this station was assigned the maximum percentage error from the other bacterial abundance. Panel D: bacterial abundance in the**
10 **microlayer and bacterial abundance in the bulk seawater from Wilson et al. (2015).**





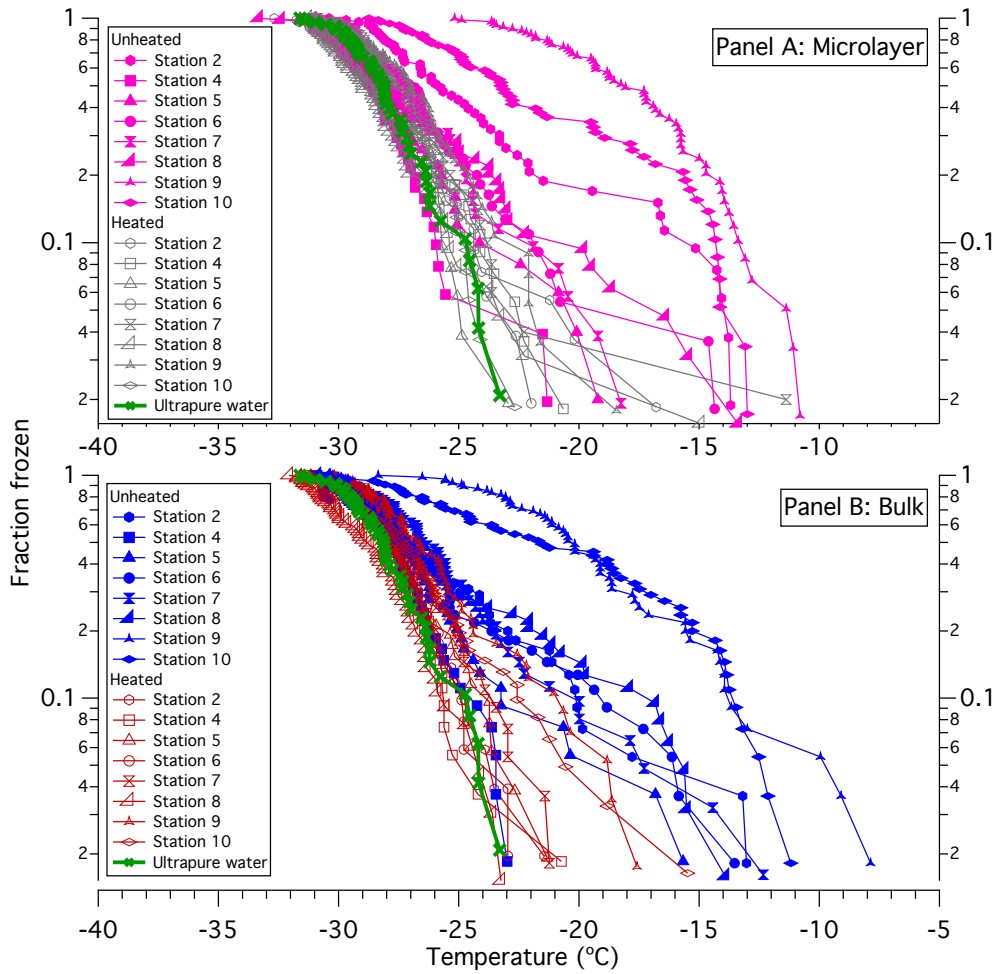

**Figure 6 - Effect of heating on the fraction frozen for unfiltered samples from microlayer (Panel A) and bulk seawater (Panel B). Each data point corresponds to one droplet freezing event, and all data have been corrected for freezing point depression.**


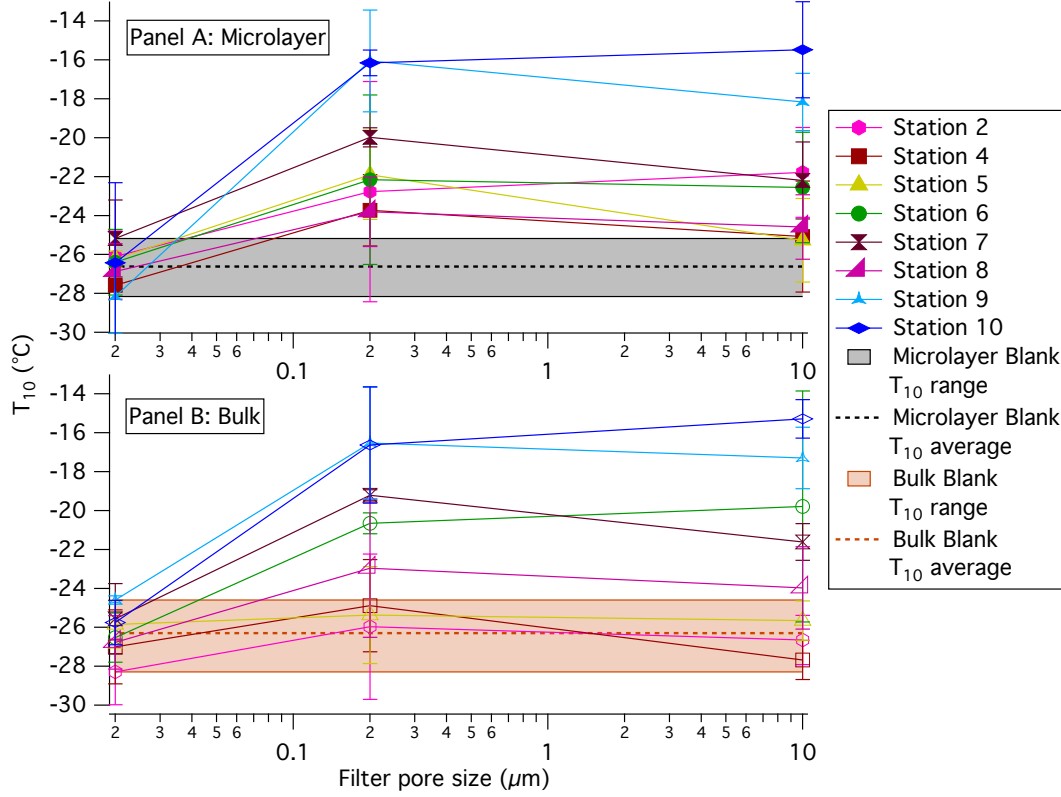

**Figure 7 - Temperature at which 10% of the droplets froze ($T_{10}$) as a function of filter pore size in microlayer samples (Panel A) and bulk seawater samples (Panel B). Filter pore sizes were 10, 0.2, and 0.02 μm. Error bars are the 95% confidence intervals of the $T_{10}$ from 3 replicate experiments. All data have been corrected for freezing point depression.**

