# Peer review of "Ice nucleating particles in Canadian Arctic sea-surface microlayer and bulk seawater"

_Atmospheric Chemistry and Physics, 2017_

## Referee Comment (RC1) · Anonymous Referee #1 · 27 May 2017

**Title:** *Ice nucleating particles in Canadian Arctic sea-surface microlayer and bulk seawater.*
**Authors:** Irish *et al.* 2017
**DOI:** 10.5194/acp-2017-384
**Journal:** Atmospheric Chemistry and Physics (ACP)

**Summary**

This study evaluates the abundance and characteristics of ice nucleating particles (INPs) in Canadian Arctic waters during the summer of 2014. Nucleation in the immersion freezing mode was quantified for microlayer and bulk (subsurface) seawater samples. Analysis of samples from eight process stations reveal that both bulk and microlayer samples contained elevated INP concentrations compared to an ultrapure water control, as well as station blanks (sample passed through a 0.02 µm filter). Contrary to previous experiments (Wilson *et al.* 2015), the results do not indicate an enrichment of INPs in the microlayer relative to the bulk samples. The concentration of INPs varies considerably between the eight geographically diverse process stations. The authors correlate INP concentrations with an array of 6 other variables (DMS concentration, bacterial and phytoplankton cell counts, temperature, pH, and salinity), finding that salinity provided the strongest and only statistically significant relationship with INP concentration. Finally, filtration and heating experiments suggest such INPs were between 0.02 µm and 0.2 µm in diameter and thermolabile, suggesting INPs were organic in composition and may have consisted of femtoplankton, cell fragments, or cell exudates/lysates.

**General Comments**

In all, this paper makes an important contribution to the field by helping to quantify the range of variability in INP concentration sourced from Arctic marine waters. The authors' effort to compare their results with other recent measurements is particularly laudable, and highlights the need for subsequent studies that contrast how oceanic variables, laboratory protocol, and sampling techniques ultimately affect measured INP concentrations. As such, this paper sets the stage nicely for further developments in the field, and therefore is well qualified for publication within ACP. Below, I offer a few questions and comments to strengthen the paper and clarify the results:

**Scientific Comments**

**1. Page 3, Lines 8 – 9:** The authors report that the Niskin bottles were washed with large amounts of seawater before sampling at 0.5 meters below the surface commenced. At the same time, the results suggest that the microlayer samples did exhibit significant INP enrichment relative to the bulk seawater, in contrast to Wilson *et al.* 2015. Were the Niskin bottles rinsed on-site in the zodiac? If so, is it possible that the organic-enriched microlayer was disturbed in the rinsing process, mixing INPs to subsurface waters and muddling the distinction between the two samples that would ordinarily exist?

**2. Page 3, Lines 33 – 35:** Heating seawater to high temperatures (100 ˚C) and gradual cooling can cause salts, especially carbonates, to precipitate from solution (e.g. Anderson, 2005; Jones 1967; Harrison et al. 1980). Were precipitates observed during the

thermodegradation heating tests? If so, have the calculations for Corrections for Freezing Temperature Depression (Section 2.4.4) taken this change in salinity/alkalinity into account? If precipitates did form and were unnoticed, then the corrected temperatures reported in Figure 4 may in fact be lower limits.

**3. Page 5, Lines 15 – 17:** A strong anticorrelation between salinity and ice nucleation efficacy was observed across the 8 process stations' samples. Studies have found that ice rafting and melting spurs cell growth (e.g. iron fertilization in the Southern Ocean (Duprat et al. 2016) and possibly phosphorus addition in the Arctic (Perrette et al. 2011)). Although there was only a moderate correlation between cell count and T10 value (or better in the microlayer – Table S2), was there a correlation between salinity and cell count? This would suggest that nutrient addition from melt water might be spurring cell growth and possibly INP production, with interesting implications for future Arctic and Greenland ice loss.

**Clarification and Technical Comments**
**1. Page 1, Line 22 – 23:** "INPs were ubiquitous in the microlayer and bulk seawater with freezing temperatures *in the immersion mode* as high as -14 °C," or something similar to indicate mode of activation.

**2. Page 2, Line 10 – 11:** "Modeling studies have also suggested that marine INPs may offset the magnitude of anthropogenic aerosol forcing by influencing cloud formation (Yun and Penner, 2013)." This is vaguely worded. How specifically do marine INPs reduce the negative anthropogenic forcing?

**3. Page 2, Line 23 – 25:** The original (Vali 1971) notation makes it more clear that INP concentration is a function of temperature: use [INP(T)] instead of [INP]. This notation more clearly denotes that INP activation is temperature-dependent.

**4. Page 16, Figure 2:** What are the temperature uncertainties on a typical data point?

**5. Page 18, Figure 4:** Since the data from Schnell (1975) and Schnell and Vali (1975) are so sparse, they would be easier to see if they were plotted over the other data.

**6. Page 18, Figure 4:** How are the reported uncertainties in [INP(T)] calculated?

**6. Supplement Pages 4-6, Figures S1–S3:** It would be helpful if the x and y axes sizes and marks (latitude and longitude labels) were consistent between the sample-site plots and the chlorophyll a plots. That way, chlorophyll concentrations at sampling sites could more easily be determined.

**Works Cited**
Anderson, R.A. (ed.), Algal Culturing Techniques, Elsevier Academic Press, Burlington, Massachusetts, 578 pp, 2005.
Duprat, L.P., Bigg, G.R. and Wilton, D.J., Enhanced Southern Ocean marine productivity due to fertilization by giant icebergs, Nature Geoscience, 2016.
Harrison, P.J., Waters, R.E. and Taylor, F.J.R., A broad spectrum artificial sea water medium for coastal and open ocean phytoplankton. Journal of Phycology, 16(1), pp.28-35, 1980.

Jones, G.: Precipitates from Autoclaved Seawater, Limnology and Oceanography, 12(1), 165–167, 1967.

Perrette, M., Yool, A., Quartly, G.D. and Popova, E.E., Near-ubiquity of ice-edge blooms in the Arctic, Biogeosciences, 8(2), p.515, 2011.

---

## Referee Comment (RC2) · Anonymous Referee #2 · 2 Jun 2017

**General Comment** Overall my comments are rather minor on this paper. It is a nice addition to the literature on the sources of ice nucleating particles from ocean seawater, and on expectations for enrichment or not in the sea surface microlayer. As detailed in my specific comments, I wonder if there is a reason to rule out non-colligative effects on freezing for explaining salinity variations, I felt it unfortunate that total organic carbon measurements were not included in order to compare with the Wilson et al. (2015, Nature) study, and I feel it would be nice to see the full influence of the heating studies on the temperature spectra of INPs. There is overall perhaps too much emphasis on 10% freezing conditions. Nevertheless, this is an excellent example of the suite of data that one might like to have when simultaneously collecting atmospheric samples over oceans.

[Figure]

Minor revisions are recommended. Specific questions/comments for potentially addressing are listed below.

**Specific Comments**

*Introduction*

Page 1, lines 37-38: "Homogeneous ice nucleation becomes increasingly important below approximately -33°C..." This statement struck me as odd. Why -33°C specifically? Use of such a value seems to beg also listing a droplet size and a time scale. In fact, there are abundant observations in the literature of supercooled water present at this temperature and down to 4 or more degrees below this.

Page 2, lines 13-14: It seems likely that the transfer to the atmosphere also remains a highly uncertain process on the basis of recent studies, although it is not a topic in this paper.

*Experimental*

Page 3, line 7: I was curious that there was no apparent pre-sterilization for microbial contamination. Does isopropanol assuredly do that?

Page 3, line 20: Should blanks be in quotes? The reason is that this cannot be a true blank. There are literature reports of sub-20 nm particles acting as INPs. I think you will refer to these as "blanks".

Page 3, line 21: Can you state a conductivity level on the DI water?

Page 4, Line 5: Did you happen to test the filters after rinsing with ultrapure water? Page 4, line 8: First, it would seem appropriate to state that the water activity correction is an average one based on fits, since uncertainties commonly occur. The authors may also wish to discuss how other elements in the seawater that induce non-colligative freezing effects might stymie this approach. So, for example, what if seawater contained AFPs?

*Results and Discussion*

Page 5, line 13: What is significant about this arbitrary $T_{10}$ value chosen? Should not correlations be checked for a range of fractions or at single temperatures? Also, were any TOC measurements made? This seems a missed opportunity to correlate with the relation suggested in Wilson et al. (2015). Page 5, line 20: Note the extra space at the end of this sentence. Also, can you rule out non-colligative freezing effects that scale with salinity? I have no reason to understand why this would be so, since I know of no such studies for seawater. It could be useful to show a plot of the relation you are discussing, in the supplemental material, if not in the main manuscript. Then it might be clear if the correlation shows any bias that could be explained by a constant "delta" on the freezing temperature.

Page 6, lines 12-13: I suspect that additional studies could also indicate which method is closer to correct, or if a new and more elaborate method might be warranted.

Page 7, lines 3-4: The conclusion made here provides a reason to show full temperature spectra for sizing analysis. Would differences stand out at certain temperatures? Or at lower levels of freezing?

Page 7, lines 7-9: I am curious if there are known things that are non-microbial or non-proteinaceous that are denatured by the heat level used. Is there an expectation that the composition of exudates would be unstable at 100°C? I do not know the answer, just asking, as O'Sullivan et al. (2015, Scientific Reports) does not suggest anything other than microbial fragments and proteins as being particularly heat sensitive.

*Summary and conclusions*

Page 7, line 18: "Biological materials" seems too broad or non-specific of a category. They are heat labile biological materials, which might imply something more (i.e., comment just above)?

*Figures*

Figure 6: Question - if only 15 to 20 drops are used, how are frozen fractions below 5

Figure 7: It might be interesting to see separate plots for each filtering size as a function of temperature, as in other plots. This would highlight if any differences occur at low freezing fractions at the warmest temperatures and how things vary with processing temperature. In that manner, the full exclusion of a role of larger particles in the bacterial size range might be better supported.

---

## Author Comment (AC1) · 24 Jul 2017

Prof. Daniel Cziczo
Co-Editor of Atmospheric Chemistry and Physics

Dear Dan,

Listed below are our responses to the comments from the reviewers of our manuscript. For clarity and visual distinction, the referee comments or questions are listed here in black and are preceded by bracketed, italicized numbers (e.g. *[1]*). Authors' responses are in red below each referee statement with matching numbers (e.g. *[A1]*).  We thank the reviewers for carefully reading our manuscript and for their helpful suggestions!

Sincerely,

Allan Bertram
Professor of Chemistry
University of British Columbia

**Anonymous Referee #1**

**Summary**
This study evaluates the abundance and characteristics of ice nucleating particles (INPs) in Canadian Arctic waters during the summer of 2014. Nucleation in the immersion freezing mode was quantified for microlayer and bulk (subsurface) seawater samples. Analysis of samples from eight process stations reveal that both bulk and microlayer samples contained elevated INP concentrations compared to an ultrapure water control, as well as station blanks (sample passed through a 0.02 μm filter). Contrary to previous experiments (Wilson et al. 2015), the results do not indicate an enrichment of INPs in the microlayer relative to the bulk samples. The concentration of INPs varies considerably between the eight geographically diverse process stations. The authors correlate INP concentrations with an array of 6 other variables (DMS concentration, bacterial and phytoplankton cell counts, temperature, pH, and salinity), finding that salinity provided the strongest and only statistically significant relationship with INP concentration. Finally, filtration and heating experiments suggest such INPs were between 0.02 μm and 0.2 μm in diameter and thermolabile, suggesting INPs were organic in composition and may have consisted of femtoplankton, cell fragments, or cell exudates/lysates.

**General Comments**
In all, this paper makes an important contribution to the field by helping to quantify the range of variability in INP concentration sourced from Arctic marine waters. The authors' effort to compare their results with other recent measurements is particularly laudable, and highlights the need for subsequent studies that contrast how oceanic variables,

laboratory protocol, and sampling techniques ultimately affect measured INP concentrations. As such, this paper sets the stage nicely for further developments in the field, and therefore is well qualified for publication within ACP. Below, I offer a few questions and comments to strengthen the paper and clarify the results:

**Scientific Comments**
*[1]* Page 3, Lines 8 – 9: The authors report that the Niskin bottles were washed with large amounts of seawater before sampling at 0.5 meters below the surface commenced. At the same time, the results suggest that the microlayer samples did exhibit significant INP enrichment relative to the bulk seawater, in contrast to Wilson et al. 2015. Were the Niskin bottles rinsed on-site in the zodiac? If so, is it possible that the organic enriched microlayer was disturbed in the rinsing process, mixing INPs to subsurface waters and muddling the distinction between the two samples that would ordinarily exist?

*[A1] To address the referee's comment more information on the how the Niskin bottles were rinsed will be provided. Specifically the following text will be added to the manuscript (Section 2.1):*

"The Niskin bottle was not cleaned with isopropanol before sampling, but the inside of the bottle was rinsed with a large amount of seawater by lowering and leaving it in the seawater with the top and bottom lids open for about a minute before sending down the messenger to close the lids for sample collection. Sampling with the Niskin bottle and the hand-held glass plate were done on opposite sides of the zodiac to minimize the effect of sampling with the Niskin bottle on the microlayer."

*[2]* Page 3, Lines 33 – 35: Heating seawater to high temperatures (100 °C) and gradual cooling can cause salts, especially carbonates, to precipitate from solution (e.g. Anderson, 2005; Jones 1967; Harrison et al. 1980). Were precipitates observed during the thermodegradation heating tests? If so, have the calculations for Corrections for Freezing Temperature Depression (Section 2.4.4) taken this change in salinity/alkalinity into account? If precipitates did form and were unnoticed, then the corrected temperatures reported in Figure 4 may in fact be lower limits.

*[A2] No precipitate was observed during the thermo degradation heating tests. Based on references suggested by the referee (Jones, 1967 and Harrison et al. 1980), if precipitate did form, the mass of the precipitate would be small compared to the total mass of dissolved material in seawater. Hence, the effect of precipitate on salinity would be small and would not significantly change the corrections for freezing point depression. Please let us know if we have misunderstood this comment.*

*[3]* Page 5, Lines 15 – 17: A strong anticorrelation between salinity and ice nucleation efficacy was observed across the 8 process stations' samples. Studies have found that ice rafting and melting spurs cell growth (e.g. iron fertilization in the Southern Ocean (Duprat et al. 2016) and possibly phosphorus addition in the Arctic (Perrette et al.

2011)). Although there was only a moderate correlation between cell count and T10 value (or better in the microlayer – Table S2), was there a correlation between salinity and cell count? This would suggest that nutrient addition from melt water might be spurring cell growth and possibly INP production, with interesting implications for future Arctic and Greenland ice loss.

*[A3] There exists a strong positive correlation between bacterial abundance and salinity in the microlayer (R = 0.76, p = 0.039). This information will be added to the revised manuscript.*

**Clarification and Technical Comments**
*[4]* Page 1, Line 22 – 23: "INPs were ubiquitous in the microlayer and bulk seawater with freezing temperatures in the immersion mode as high as -14 °C," or something similar to indicate mode of activation.

*[A4] The authors will change the wording here as suggested.*

*[5]* Page 2, Line 10 – 11: "Modeling studies have also suggested that marine INPs may offset the magnitude of anthropogenic aerosol forcing by influencing cloud formation (Yun and Penner, 2013)." This is vaguely worded. How specifically do marine INPs reduce the negative anthropogenic forcing?

*[A5] The authors will be more specific in wording this sentence in the revised manuscript.*

*[6]* Page 3, Line 23 – 25: The original (Vali 1971) notation makes it more clear that INP concentration is a function of temperature: use [INP(T)] instead of [INP]. This notation more clearly denotes that INP activation is temperature-dependent.

*[A6] The term [INP] will be replaced with [INP(T)].*

*[7]* Page 16, Figure 2: What are the temperature uncertainties on a typical data point?

*[A7] The figure caption will be updated to report the uncertainty in temperature (±0.3 °C).*

*[8]* Page 18, Figure 4: Since the data from Schnell (1975) and Schnell and Vali (1975) are so sparse, they would be easier to see if they were plotted over the other data.

*[A8] The authors will make sure this figure is updated according to this suggestion.*

*[9]* Page 18, Figure 4: How are the reported uncertainties in [INP(T)] calculated?

*[A9] The method of calculating uncertainties in Figure 4 will be added to the manuscript to address the referee's comments.*

[10] Supplement Pages 4-6, Figures S1–S3: It would be helpful if the x and y axes sizes and marks (latitude and longitude labels) were consistent between the sample-site plots and the chlorophyll a plots. That way, chlorophyll concentrations at sampling sites could more easily be determined.

*[A10] As suggested, the x and y axes sizes and marks will be adjusted so they are consistent between the sample-site plots and the chlorophyll a plots.*

**Works Cited**

Anderson, R.A. (ed.), Algal Culturing Techniques, Elsevier Academic Press, Burlington, Massachusetts, 578 pp, 2005.

Duprat, L.P., Bigg, G.R. and Wilton, D.J., Enhanced Southern Ocean marine productivity due to fertilization by giant icebergs, Nature Geoscience, 2016.

Harrison, P.J., Waters, R.E. and Taylor, F.J.R., A broad spectrum artificial sea water medium for coastal and open ocean phytoplankton. Journal of Phycology, 16(1), pp.28-35, 1980.

Jones, G.: Precipitates from Autoclaved Seawater, Limnology and Oceanography, 12(1), 165– 167, 1967.

Perrette, M., Yool, A., Quartly, G.D. and Popova, E.E., Near-ubiquity of ice-edge blooms in the Arctic, Biogeosciences, 8(2), p.515, 2011.

**Anonymous Referee #2**

**General Comment**

Overall my comments are rather minor on this paper. It is a nice addition to the literature on the sources of ice nucleating particles from ocean seawater, and on expectations for enrichment or not in the sea surface microlayer. As detailed in my specific comments, I wonder if there is a reason to rule out non-colligative effects on freezing for explaining salinity variations, I felt it unfortunate that total organic carbon measurements were not included in order to compare with the Wilson et al. (2015, Nature) study, and I feel it would be nice to see the full influence of the heating studies on the temperature spectra of INPs. There is overall perhaps too much emphasis on 10% freezing conditions. Nevertheless, this is an excellent example of the suite of data that one might like to have when simultaneously collecting atmospheric samples over oceans. Minor revisions are recommended. Specific questions/comments for potentially addressing are listed below.

**Specific Comments**
*Introduction*
[11] Page 1, lines 37-38: "Homogeneous ice nucleation becomes increasingly important below approximately -33∘C. . ." This statement struck me as odd. Why -33∘C specifically? Use of such a value seems to beg also listing a droplet size and a time scale. In fact, there are abundant observations in the literature of supercooled water present at this temperature and down to 4 or more degrees below this.

*[A11] To address the referee's comments we will add justification for the choice of -33 °C in the revised manuscript.*

*[12]* Page 2, lines 13-14: It seems likely that the transfer to the atmosphere also remains a highly uncertain process on the basis of recent studies, although it is not a topic in this paper.

*[A12] To address the referee's comments, this sentence will be modified to make it clear that the transfer to the atmosphere is also uncertain.*

*Experimental*
*[13]* Page 3, line 7: I was curious that there was no apparent pre-sterilization for microbial contamination. Does isopropanol assuredly do that?

*[A13] Isopropanol has been used in previous pre-sterilisation protocols (Csuros, M., Environmental Sampling and Analysis for technicians, Lewis Publishers, NY, 1994). This will be made clear in the revised manuscript.*

*[14]* Page 3, line 20: Should blanks be in quotes? The reason is that this cannot be a true blank. There are literature reports of sub-20 nm particles acting as INPs. I think you will refer to these as "blanks".

*[A14] As suggested we will use quotes when referring to blanks and also point out that the "blanks" may still contain sub-20 nm particles that can act as INPs.*

*[15]* Page 3, line 21: Can you state a conductivity level on the DI water?

*[A15] In the revised manuscript the conductivity level will be stated.*

*[16]* Page 4, Line 5: Did you happen to test the filters after rinsing with ultrapure water?

*[A16] The filters were not tested after rinsing with ultrapure water. Is the referee concerned that the filters contain INPs?*

*[17]* Page 4, line 8: First, it would seem appropriate to state that the water activity correction is an average one based on fits, since uncertainties commonly occur. The authors may also wish to discuss how other elements in the seawater that induce non-colligative freezing effects might stymie this approach. So, for example, what if seawater contained AFPs?

*[A17] To address the referee's comments in the revised manuscript we will point out the approximations in the water activity correction and also point out that the water activity correction does not consider non-colligative freezing effects, although non-colligative freezing effects have not been observed in previous immersion studies with sodium chloride solutions or seawater.*

*Results and Discussion*

*[18]* Page 5, line 13: What is significant about this arbitrary T10 value chosen? Should not correlations be checked for a range of fractions or at single temperatures? Also, were any TOC measurements made? This seems a missed opportunity to correlate with the relation suggested in Wilson et al. (2015).

*[A18] To address the referee's comment, in the revised manuscript we will check for correlations at an additional fraction, specifically $T_{50}$. This information will be added to the supplement to limit the size of the main text. Unfortunately, we do not have reliable measurements of TOC from the cruise.*

*[19]* Page 5, line 20: Note the extra space at the end of this sentence. Also, can you rule out non-colligative freezing effects that scale with salinity? I have no reason to understand why this would be so, since I know of no such studies for seawater. It could be useful to show a plot of the relation you are discussing, in the supplemental material, if not in the main manuscript. Then it might be clear if the correlation shows any bias that could be explained by a constant "delta" on the freezing temperature.

*[A19] We can not rule out non-colligative freezing effects, but on the other hand, non-colligative freezing effects have not been observed in previous immersion freezing studies with sodium chloride solutions (Alpert et al., 2011a, 2011b; Knopf et al., 2011; Zobrist et al., 2008) or seawater (Wilson et al., 2015). To address the referee's comments, this information will be added to the revised manuscript. In addition, correlation plots between $T_{10}$ and the variables investigated will be added in the supplement.*

*[20]* Page 6, lines 12-13: I suspect that additional studies could also indicate which method is closer to correct, or if a new and more elaborate method might be warranted.

*[A20] We are not sure if the referee would like us to add additional text about this point.*

*[21]* Page 7, lines 3-4: The conclusion made here provides a reason to show full temperature spectra for sizing analysis. Would differences stand out at certain temperatures? Or at lower levels of freezing?

*[A21] To address the referee's comments, we will add a figure to the supplement that shows the full temperature spectra for the sizing analysis.*

*[22]* Page 7, lines 7-9: I am curious if there are known things that are non-microbial or nonproteinaceous that are denatured by the heat level used. Is there an expectation that the composition of exudates would be unstable at 100◦C? I do not know the answer, just asking, as O'Sullivan et al. (2015, Scientific Reports) does not suggest anything other than microbial fragments and proteins as being particularly heat sensitive.

*[A22] In our discussion where we suggested exudates as the possible source of INPs we assumed that exudates are unstable/denature at 100 °C. However, we recognise that this is not a certainty and we will change the sentence to reflect this point. We are not aware of studies that have investigated the stability of exudates to heat.*

*Summary and conclusions*
*[23]* Page 7, line 18: "Biological materials" seems too broad or non-specific of a category. They are heat labile biological materials, which might imply something more (i.e., comment just above)?

*[A23] In the revised manuscript, the term biological materials will be modified to heat labile biological materials.*

*Figures*
*[24]* Figure 6: Question - if only 15 to 20 drops are used, how are frozen fractions below 5

*[A24] 15 to 20 drops were used in a single freezing experiment, but for each sample, freezing experiments were performed 3 times, resulting in 45-60 freezing events per sample. This point will be made clear in the revised manuscript.*

*[25]* Figure 7: It might be interesting to see separate plots for each filtering size as a function of temperature, as in other plots. This would highlight if any differences occur at low freezing fractions at the warmest temperatures and how things vary with processing temperature. In that manner, the full exclusion of a role of larger particles in the bacterial size range might be better supported.

*[A25] See response [A21].*

---

## Author Response (AR1)

Prof. Daniel Cziczo
Co-Editor of Atmospheric Chemistry and Physics

Dear Dan,

Listed below are our responses to the comments from the reviewers of our manuscript. For clarity and visual distinction, the referee comments or questions are listed here in black and are preceded by bracketed, italicized numbers (e.g. *[1]*). Authors' responses are in red below each referee statement with matching numbers (e.g. *[A1]*). We thank the reviewers for carefully reading our manuscript and for their helpful suggestions!

Sincerely,

Allan Bertram
Professor of Chemistry
University of British Columbia

**Anonymous Referee #1**

**Summary**
This study evaluates the abundance and characteristics of ice nucleating particles (INPs) in Canadian Arctic waters during the summer of 2014. Nucleation in the immersion freezing mode was quantified for microlayer and bulk (subsurface) seawater samples. Analysis of samples from eight process stations reveal that both bulk and microlayer samples contained elevated INP concentrations compared to an ultrapure water control, as well as station blanks (sample passed through a 0.02 μm filter). Contrary to previous experiments (Wilson et al. 2015), the results do not indicate an enrichment of INPs in the microlayer relative to the bulk samples. The concentration of INPs varies considerably between the eight geographically diverse process stations. The authors correlate INP concentrations with an array of 6 other variables (DMS concentration, bacterial and phytoplankton cell counts, temperature, pH, and salinity), finding that salinity provided the strongest and only statistically significant relationship with INP concentration. Finally, filtration and heating experiments suggest such INPs were between 0.02 μm and 0.2 μm in diameter and thermolabile, suggesting INPs were organic in composition and may have consisted of femtoplankton, cell fragments, or cell exudates/lysates.

**General Comments**
In all, this paper makes an important contribution to the field by helping to quantify the range of variability in INP concentration sourced from Arctic marine waters. The authors' effort to compare their results with other recent measurements is particularly laudable, and highlights the need for subsequent studies that contrast how oceanic variables, laboratory protocol, and sampling techniques ultimately affect measured INP concentrations. As such, this paper sets the stage nicely for further developments in the field, and therefore is well qualified for publication within ACP. Below, I offer a few questions and comments to strengthen the paper and clarify the results:

**Scientific Comments**

*[1] Page 3, Lines 8 – 9: The authors report that the Niskin bottles were washed with large amounts of seawater before sampling at 0.5 meters below the surface commenced. At the same time, the results suggest that the microlayer samples did exhibit significant INP enrichment relative to the bulk seawater, in contrast to Wilson et al. 2015. Were the Niskin bottles rinsed on-site in the zodiac? If so, is it possible that the organic enriched microlayer was disturbed in the rinsing process, mixing INPs to subsurface waters and muddling the distinction between the two samples that would ordinarily exist?*

*[A1] To address the referee's comment we provided more information on the how the Niskin bottles were rinsed. Specifically the following text was added to the manuscript (Section 2.1):*

*"The Niskin bottle was not cleaned with isopropanol before sampling, but the inside of the bottle was rinsed with a large amount of seawater by lowering and leaving it in the seawater with the top and bottom lids open for about a minute before sending down the messenger to close the lids for sample collection. Sampling with the Niskin bottle and the hand-held glass plate were done on opposite sides of the zodiac to minimize the effect of sampling with the Niskin bottle on the microlayer."*

*[2] Page 3, Lines 33 – 35: Heating seawater to high temperatures (100 ˚C) and gradual cooling can cause salts, especially carbonates, to precipitate from solution (e.g. Anderson, 2005; Jones 1967; Harrison et al. 1980). Were precipitates observed during the thermodegradation heating tests? If so, have the calculations for Corrections for Freezing Temperature Depression (Section 2.4.4) taken this change in salinity/alkalinity into account? If precipitates did form and were unnoticed, then the corrected temperatures reported in Figure 4 may in fact be lower limits.*

*[A2] No precipitate was observed during the thermo degradation heating tests. Based on references suggested by the referee (Jones, 1967 and Harrison et al. 1980), if precipitate did form, the mass of the precipitate would be small compared to the total mass of dissolved material in seawater. Hence, the effect of precipitate on salinity would be small and would not significantly change the corrections for freezing point depression. Please let us know if we have misunderstood this comment.*

*[3] Page 5, Lines 15 – 17: A strong anticorrelation between salinity and ice nucleation efficacy was observed across the 8 process stations' samples. Studies have found that ice rafting and melting spurs cell growth (e.g. iron fertilization in the Southern Ocean (Duprat et al. 2016) and possibly phosphorus addition in the Arctic (Perrette et al. 2011)). Although there was only a moderate correlation between cell count and T10 value (or better in the microlayer – Table S2), was there a correlation between salinity and cell count? This would suggest that nutrient addition from melt water might be spurring cell growth and possibly INP production, with interesting implications for future Arctic and Greenland ice loss.*

*[A3] There exists a strong positive correlation between bacterial abundance and salinity in the microlayer (R = 0.76, p = 0.039). To address the referee's comment the following text has been added to Section 3.1:*

*"Also interesting, a strong positive correlation was observed between salinity and bacterial abundance (R = 0.76, p = 0.039). Consistent with these results, Galgani et al. (2016) observed a higher concentration of bacteria in the open sea (which had a higher salinity) compared to melt ponds (which had a lower salinity)."*

**Clarification and Technical Comments**

*[4]* Page 1, Line 22 – 23: "INPs were ubiquitous in the microlayer and bulk seawater with freezing temperatures in the immersion mode as high as -14 °C," or something similar to indicate mode of activation.

*[A4] The authors changed the wording in the abstract as suggested.*

*[5]* Page 2, Line 10 – 11: "Modeling studies have also suggested that marine INPs may offset the magnitude of anthropogenic aerosol forcing by influencing cloud formation (Yun and Penner, 2013)." This is vaguely worded. How specifically do marine INPs reduce the negative anthropogenic forcing?

*[A5] The authors have been more specific in wording this sentence in the revised manuscript. The following is the revised sentence added in Section 1:*

*"Modelling studies show that natural marine INPs may contribute to more ice formation in mixed-phase clouds, thereby reducing the magnitude of the total top-of-atmosphere anthropogenic aerosol forcing by as much as 0.3 W/m$^2$ (Yun and Penner, 2013)."*

*[6]* Page 3, Line 23 – 25: The original (Vali 1971) notation makes it more clear that INP concentration is a function of temperature: use [INP(T)] instead of [INP]. This notation more clearly denotes that INP activation is temperature-dependent.

*[A6] The term [INP] has been replaced with [INP(T)] throughout the manuscript.*

*[7]* Page 16, Figure 2: What are the temperature uncertainties on a typical data point?

*[A7] The caption for Figure 2 has been updated to report the uncertainty in temperature (±0.3 °C).*

*[8]* Page 18, Figure 4: Since the data from Schnell (1975) and Schnell and Vali (1975) are so sparse, they would be easier to see if they were plotted over the other data.

*[A8] The authors have updated Figure 4 according to this suggestion.*

*[9]* Page 18, Figure 4: How are the reported uncertainties in [INP(T)] calculated?

*[A9] In the original manuscript the upper and lower limits for [INP(T)] values were determined by assuming individual droplets in the freezing experiments could contain multiple INPs and only one INP, respectively, and the reported [INP(T)] values were midpoints between these upper and lower limits. In hindsight, this was not clear in the submitted manuscript nor was this the most appropriate way to describe our experimental data. In the revised manuscript, reported [INP(T)] values are calculated by assuming individual droplets in the freezing experiments contain multiple INPs, consistent with the discussion in Section 2.2.1. Upper and lower limits to [INP(T)] were calculated based on the limited number of nucleation events observed in the freezing experiments (Koop et al. (1997)). These upper and lower limits take into account the statistical uncertainty in the freezing experiments. The method used to calculate the uncertainties has been added to the caption of Figure 4 to make this clear.*
*Fortunately, the conclusions in our manuscript are not sensitive to the method we used to calculate [INP(T)] values or their uncertainties. Nevertheless, the method used to calculate*

*[INP(T)] values and their uncertainties in the revised manuscript is now clear and most appropriate.*

[10] Supplement Pages 4-6, Figures S1–S3: It would be helpful if the x and y axes sizes and marks (latitude and longitude labels) were consistent between the sample-site plots and the chlorophyll a plots. That way, chlorophyll concentrations at sampling sites could more easily be determined.

*[A10] As suggested, the x and y axes sizes and marks in these figures have been adjusted so they are consistent between the sample-site plots and the chlorophyll a plots (see Supplement).*

**Works Cited**

Anderson, R.A. (ed.), Algal Culturing Techniques, Elsevier Academic Press, Burlington, Massachusetts, 578 pp, 2005.

Duprat, L.P., Bigg, G.R. and Wilton, D.J., Enhanced Southern Ocean marine productivity due to fertilization by giant icebergs, Nature Geoscience, 2016.

Harrison, P.J., Waters, R.E. and Taylor, F.J.R., A broad spectrum artificial sea water medium for coastal and open ocean phytoplankton. Journal of Phycology, 16(1), pp.28-35, 1980.

Jones, G.: Precipitates from Autoclaved Seawater, Limnology and Oceanography, 12(1), 165–167, 1967.

Perrette, M., Yool, A., Quartly, G.D. and Popova, E.E., Near-ubiquity of ice-edge blooms in the Arctic, Biogeosciences, 8(2), p.515, 2011.

**Anonymous Referee #2**

**General Comment**

Overall my comments are rather minor on this paper. It is a nice addition to the literature on the sources of ice nucleating particles from ocean seawater, and on expectations for enrichment or not in the sea surface microlayer. As detailed in my specific comments, I wonder if there is a reason to rule out non-colligative effects on freezing for explaining salinity variations, I felt it unfortunate that total organic carbon measurements were not included in order to compare with the Wilson et al. (2015, Nature) study, and I feel it would be nice to see the full influence of the heating studies on the temperature spectra of INPs. There is overall perhaps too much emphasis on 10% freezing conditions. Nevertheless, this is an excellent example of the suite of data that one might like to have when simultaneously collecting atmospheric samples over oceans. Minor revisions are recommended. Specific questions/comments for potentially addressing are listed below.

**Specific Comments**

*Introduction*

[11] Page 1, lines 37-38: "Homogeneous ice nucleation becomes increasingly important below approximately -33◦C. . ." This statement struck me as odd. Why -33◦C specifi- cally? Use of such a value seems to beg also listing a droplet size and a time scale. In fact, there are abundant observations in the literature of supercooled water present at this temperature and down to 4 or more degrees below this.

*[A11] To address the referee's comments we have added the justification for the choice of -33 °C in the revised manuscript (Section 1). The revised sentence is as follows:*

*"Homogeneous ice nucleation becomes increasingly important below approximately -33 °C for typical cloud sizes and atmospheric cooling rates (Herbert et al., 2015; Koop and Murray, 2016), but INPs can trigger ice formation in clouds at higher temperatures."*

[12] Page 2, lines 13-14: It seems likely that the transfer to the atmosphere also remains a highly uncertain process on the basis of recent studies, although it is not a topic in this paper.

*[A12] To address the referee's comments, this sentence has been modified to make it clear that the transfer to the atmosphere is also uncertain. The following is the revised sentence in Section 1:*

*"Nevertheless, our current understanding of the properties, concentrations, and spatial and temporal distributions of INPs in the microlayer and bulk seawater, as well as their transfer to the atmosphere, remains limited, leading to uncertainties when predicting their impacts on climate and the hydrological cycle."*

*Experimental*
[13] Page 3, line 7: I was curious that there was no apparent pre-sterilization for microbial contamination. Does isopropanol assuredly do that?

*[A13] Isopropanol has been used in previous pre-sterilisation protocols (Csuros, M., Environmental Sampling and Analysis for technicians, Lewis Publishers, NY, 1994). This has been made clear in the revised manuscript (Section 2.1).*

[14] Page 3, line 20: Should blanks be in quotes? The reason is that this cannot be a true blank. There are literature reports of sub-20 nm particles acting as INPs. I think you will refer to these as "blanks".

*[A14] As suggested we have used quotes when referring to blanks throughout the manuscript and also point out that the "blanks" may still contain sub-20 nm particles that can act as INPs. Specifically, we have added the following text to Section 3.1:*

*"The "blanks" may still contain some INPs, since some particles < 0.02 μm in diameter can act as INPs (Dreischmeier et al., 2017; O'Sullivan et al., 2015)."*

[15] Page 3, line 21: Can you state a conductivity level on the DI water?

*[A15] In the revised manuscript the conductivity level has been stated in Section 2.2.1.*

[16] Page 4, Line 5: Did you happen to test the filters after rinsing with ultrapure water?

*[A16] The filters were not tested after rinsing with ultrapure water. Is the referee concerned that the filters contain INPs?*

[17] Page 4, line 8: First, it would seem appropriate to state that the water activity correction is an average one based on fits, since uncertainties commonly occur. The authors may also wish to discuss how other elements in the seawater that induce non-colligative freezing effects might stymie this approach. So, for example, what if seawater contained AFPs?

*[A17] To address the referee's comments in the revised manuscript we have pointed out the approximations in the water activity correction in Section 2.2.4 by using the following sentence:*

*"The freezing temperature correction was calculated using the median freezing temperature of each sample and then applied to the rest of the droplet freezing temperatures within that sample."*

*We also pointed out that the water activity correction does not consider non-colligative freezing effects, although non-colligative freezing effects have not been observed in previous immersion studies with sodium chloride solutions or seawater. The following text was added to the end of Section 2.2.4:*

*"The water activity corrections do not consider non-colligative effects; however, non-colligative effects have not been observed in previous immersion freezing studies with sodium chloride solutions (Alpert et al., 2011a, 2011b; Knopf et al., 2011; Zobrist et al., 2008) or seawater (Wilson et al., 2015)."*

*Results and Discussion*
[18] Page 5, line 13: What is significant about this arbitrary T10 value chosen? Should not correlations be checked for a range of fractions or at single temperatures? Also, were any TOC measurements made? This seems a missed opportunity to correlate with the relation suggested in Wilson et al. (2015).

*[A18] Unfortunately, we did not have reliable measurements of TOC from the cruise. To address the referee's comment, in the revised manuscript we checked for correlations at an additional fraction, specifically $T_{50}$. This information was indicated in the main text with the following sentence:*

*"A similar trend was observed for $T_{50}$-values, where $T_{50}$ represents the temperatures at which 50% of droplets had frozen (Table S2)."*

[19] Page 5, line 20: Note the extra space at the end of this sentence. Also, can you rule out non-colligative freezing effects that scale with salinity? I have no reason to understand why this would be so, since I know of no such studies for seawater. It could be useful to show a plot of the relation you are discussing, in the supplemental material, if not in the main manuscript. Then it might be clear if the correlation shows any bias that could be explained by a constant "delta" on the freezing temperature.

*[A19] We can not rule out non-colligative freezing effects, but on the other hand, non-colligative freezing effects have not been observed in previous immersion freezing studies with sodium chloride solutions (Alpert et al., 2011a, 2011b; Knopf et al., 2011; Zobrist et al., 2008) or seawater (Wilson et al., 2015). To address the referee's comment, the following information has been added to Section 3.1:*

*"Another possible explanation for the strong negative correlation between salinity and freezing temperatures is a non-colligative effect not accounted for in the corrections for freezing temperature depression discussed in Section 2.2.4. However, as mentioned in Section 2.2.4, non-colligative effects have not been observed in previous immersion freezing studies with sodium chloride solutions (Alpert et al., 2011a, 2011b; Knopf et al., 2011; Zobrist et al., 2008) or seawater (Wilson et al., 2015)."*

*In addition, correlation plots between $T_{10}$ and the variables investigated have been added in the supplement (Figure S1).*

*[20]* Page 6, lines 12-13: I suspect that additional studies could also indicate which method is closer to correct, or if a new and more elaborate method might be warranted.

*[A20] We are not sure if the referee would like us to add additional text about this point.*

*[21]* Page 7, lines 3-4: The conclusion made here provides a reason to show full temperature spectra for sizing analysis. Would differences stand out at certain temperatures? Or at lower levels of freezing?

*[A21] To address the referee's comments, we have added a figure to the supplement (Figure S5) that shows the full temperature spectra for the sizing analysis.*

*[22]* Page 7, lines 7-9: I am curious if there are known things that are non-microbial or nonproteinaceous that are denatured by the heat level used. Is there an expectation that the composition of exudates would be unstable at 100∘C? I do not know the answer, just asking, as O'Sullivan et al. (2015, Scientific Reports) does not suggest anything other than microbial fragments and proteins as being particularly heat sensitive.

*[A22] We are not aware of studies that have investigated the stability of exudates to heat. In our discussion where we suggested exudates as the possible source of INPs we assumed that exudates are unstable/denature at 100 °C. However, we recognised that this is not a certainty and we have changed this sentence to reflect this point in Section 3.2.2. Below is the original sentence followed by the modified sentence:*

*"Potential sources of the INPs observed in this study include ultramicrobacteria, viruses, phytoplankton exudates, or bacteria exudates, all of which could be denatured by heat, but are less than 0.2 µm in size (Ladino et al., 2016; Wilson et al., 2015)."*

*Modified sentence:*

*"Potential sources of the INPs observed in this study include ultramicrobacteria, viruses, phytoplankton exudates, or bacteria exudates (Ladino et al., 2016; Wilson et al., 2015)."*

*Summary and conclusions*
*[23]* Page 7, line 18: "Biological materials" seems too broad or non-specific of a category. They are heat labile biological materials, which might imply something more (i.e., comment just above)?

*[A23] In the revised manuscript, the term biological material has been modified to heat labile biological materials throughout the manuscript.*

*Figures*
*[24]* Figure 6: Question - if only 15 to 20 drops are used, how are frozen fractions below 5

*[A24] 15 to 20 drops were used in a single freezing experiment, but for each sample, freezing experiments were performed 3 times, resulting in 45-60 freezing events per sample. This point*

*has been made clear in the revised manuscript with the following sentence added to the caption of Figure 2:*

*"Each set of line and markers represents the results for 3 repeat experiments of a sample or "blank", adding up to a total of between 45 to 60 freezing events in each set."*

*[25]* Figure 7: It might be interesting to see separate plots for each filtering size as a function of temperature, as in other plots. This would highlight if any differences occur at low freezing fractions at the warmest temperatures and how things vary with processing temperature. In that manner, the full exclusion of a role of larger particles in the bacterial size range might be better supported.

*[A25] See response [A21].*

[revised manuscript text omitted]

Victoria Irish 2017-7-26 3:15 PM

**Comment [32]:** Comment [19]

[Figure]

**Figure S2 - Sample locations and monthly average chlorophyll *a* concentrations for sampling during the current study. Chlorophyll *a* concentrations were obtained from the NASA Ocean Biology Distributed Active Archive Centre (OB.DAAC).**

[Figure]

**Figure S3 - Sample locations and monthly average chlorophyll *a* concentrations for sampling during the Wilson et al. (2015) study in the Arctic. Chlorophyll *a* concentrations were obtained from the NASA Ocean Biology Distributed Active Archive Centre (OB.DAAC).**

[Figure]

**Figure S4 - Sample locations and monthly average chlorophyll *a* concentrations for sampling during the Wilson et al. (2015) study in the Atlantic. Chlorophyll *a* concentrations were obtained from the NASA Ocean Biology Distributed Active Archive Centre (OB.DAAC).**

Victoria Irish 2017-7-26 3:13 PM
**Comment [33]:** Comment [10]

[Figure]

Victoria Irish 2017-7-26 3:13 PM
**Comment [34]:** Comment [21] and [25]

**Figure S5** - Plots of the fraction of droplets frozen (in the immersion mode) versus temperature for samples filtered with 10 μm, 0.2 μm and 0.02 μm filters. Panel A and Panel B correspond to the microlayer and bulk seawater, respectively. Each set of line and markers represents results for 3 repeat experiments of each sample or "blank", adding up to a total of between 45 to 60 freezing events in each set. All microlayer and bulk seawater freezing points have been corrected for freezing point depression to account for dissolved salts in seawater (Section 2.2.4). The uncertainty in temperature is not shown but is ± 0.3 °C.

Victoria Irish 2017-7-26 3:15 PM
**Comment [35]:** Comment [24]

Victoria Irish 2017-7-26 3:15 PM
**Comment [36]:** Comment [7]